# Plasma metabolomic analysis indicates flavonoids and sorbic acid are associated with incident diabetes: A nested case-control study among Women's Interagency HIV Study participants

**Elaine A. Yu[1]⊚, José O. Alemán[ID][2]⊚, Donald R. Hoover[3], Qiuhu Shi[4], Michael Verano[2], Kathryn Anastos[5], Phyllis C. Tien[6,7], Anjali Sharma[5], Ani Kardashian[8], Mardge H. Cohen[9], Elizabeth T. Golub[10], Katherine G. Michel[11], Deborah R. Gustafson[12], Marshall J. Glesby[ID][13]***

1 Rollins School of Public Health, Emory University, Atlanta, Georgia, United States of America, 2 Laboratory of Translational Obesity Research, New York University Grossman School of Medicine, New York, New York, United States of America, 3 Department of Statistics and Biostatistics, Institute for Health, Health Care Policy and Aging Research, Rutgers University, New Brunswick, New Jersey, United States of America, 4 New York Medical College, Valhalla, New York, United States of America, 5 Montefiore Medical Center, Bronx, New York, United States of America, 6 University of California, San Francisco, California, United States of America, 7 Department of Veterans Affairs Medical Center, San Francisco, California, United States of America, 8 University of Southern California, Los Angeles, California, United States of America, 9 Cook County Health & Hospitals System and Rush University, Chicago, Illinois, United States of America, 10 Johns Hopkins Bloomberg School of Public Health, Baltimore, Maryland, United States of America, 11 Georgetown University School of Medicine, District of Columbia, United States of America, 12 State University of New York Downstate Health Sciences University, New York, New York, United States of America, 13 Division of Infectious Diseases, Weill Cornell Medicine, New York, New York, United States of America

⊚ These authors contributed equally to this work.
* mag2005@med.cornell.edu

## Abstract

### Introduction

Lifestyle improvements are key modifiable risk factors for Type 2 diabetes mellitus (DM) however specific influences of biologically active dietary metabolites remain unclear. Our objective was to compare non-targeted plasma metabolomic profiles of women with versus without confirmed incident DM. We focused on three lipid classes (fatty acyls, prenol lipids, polyketides).

### Materials and methods

Fifty DM cases and 100 individually matched control participants (80% with human immuno-deficiency virus [HIV]) were enrolled in a case-control study nested within the Women's Interagency HIV Study. Stored blood samples (1–2 years prior to DM diagnosis among cases; at the corresponding timepoint among matched controls) were assayed in triplicate for metabolomics. Time-of-flight liquid chromatography mass spectrometry with dual electrospray ionization modes was utilized. We considered 743 metabolomic features in a

**Data Availability Statement:** All relevant data are within the paper and its Supporting information files.

**Funding:** The WIHS is funded by the National Institute of Allergy and Infectious Diseases (UO1-AI-35004, U01-AI-31834, U01-AI-34994, U01-AI-34989, U01-AI-34993, and U01-AI-42590) and by the Eunice Kennedy Shriver National Institute of Child Health and Human Development (UO1-HD-32632). The study is co-funded by the National Cancer Institute, the National Institute on Drug Abuse, and the National Institute on Deafness and Other Communication Disorders. Funding is also provided by the National Center for Research Resources (UCSF-CTSI Grant Number UL1 RR024131). This manuscript was also partially supported by NIDDK K08-DK117064 (J.O.A.). The funders had no role in study design, data collection and analysis, decision to publish, or preparation of the manuscript. There was no additional external funding received for this study.

**Competing interests:** The authors have declared that no competing interests exist.

two-stage feature selection approach with conditional logistic regression models that accounted for matching strata.

## Results

Seven features differed by DM case status (all false discovery rate-adjusted q<0.05). Three flavonoids (two flavanones, one isoflavone) were respectively associated with lower odds of DM (all q<0.05), and sorbic acid was associated with greater odds of DM (all q<0.05).

## Conclusion

Flavonoids were associated with lower odds of incident DM while sorbic acid was associated with greater odds of incident DM.

## Introduction

Diabetes mellitus (DM) is associated with an increasingly heavy burden of disease globally [1,2], including among people with human immunodeficiency virus (HIV) [3,4]. Over the last three decades, the number of people with DM more than doubled from 211 million in 1990 to 476 million in 2017 [1]. This increase largely reflects the growing number of people with Type 2 diabetes mellitus (T2DM), which also accounts for most DM cases [1]. A major obstacle to reducing T2DM incidence, prevalence, and mortality is increasing the effectiveness of prevention strategies, including through an improved understanding of modifiable risk factors [5] in diverse phenotypic subgroups.

Lifestyle modifications, including healthier dietary patterns with more fruits and vegetables and fewer processed foods, are key prevention recommendations for reducing the risk of T2DM [2]. Despite a large literature regarding specific diets [6] and nutrients [7] in association with diabetes outcomes, findings across some previous studies are inconsistent [8]. It remains a challenge to account for the extensive inter- and intra-individual heterogeneity in consumption patterns, nutritional requirements, dietary responses (e.g., nutrient absorption) [9] as well as the roles of non-nutrients and other dietary components [10]. Evaluation of dietary interventions, particularly long-term adherence, is a major obstacle. Circulating biomarkers of dietary intake could circumvent these issues and potentially serve as improved metrics of specific biologically-active metabolites and earlier predictors of long-term metabolic health [11–13].

Metabolomics can provide high-throughput, comprehensive, and relatively non-biased examination of low molecular weight metabolites [14]. Metabolomic data have the potential to characterize overall dietary intake and to identify earlier, modifiable dietary risk factors for DM [14]. Branched-chain amino acids and sphingolipids have been extensively evaluated in the context of insulin resistance and DM [15,16]. In a recent study among Women's Interagency HIV Study (WIHS) participants, cholesteryl esters, diacylglycerols, lysophosphatidylcholines, phosphatidylcholines, and phosphatidylethanolamines were associated with diabetes risk [17].

This individually matched nested case-control study compared non-targeted plasma metabolomic profiles among women with versus without confirmed, incident DM. We evaluated lipids and lipid classes that represent potential dietary modifiable risk factors of DM. Specifically, our focus was on three classes of lipids (fatty acyls, prenol lipids, polyketides) [18].

## Materials and methods

### Study participants

WIHS was a multicenter prospective cohort study among U.S. women with HIV and women without HIV who had similar risk behaviors as HIV-seropositive women [19,20]. WIHS merged with the Multicenter AIDS Cohort Study (MACS) in 2019 to form the MACS/WIHS Combined Cohort Study [21]. In WIHS, HIV-seronegative women were enrolled based upon having similar risk behaviors as HIV-seropositive women [19,20]. This study included data collected from 3,772 women enrolled at six WIHS consortia (Bronx/Manhattan, NY; Brooklyn, NY; Los Angeles/Southern California/Hawaii; San Francisco/Bay Area, CA; Chicago, IL; Washington, DC) [19]. This nested-case control study included 50 cases and 100 matched controls in the final analytic dataset (S1 Fig).

### Data collection

As part of the parent cohort study, participants completed in study visits every six months from October 2000 to April 2008. At baseline and at each semi-annual follow-up visit, women completed questionnaires regarding self-reported sociodemographics, behavioral risk and lifestyle factors. During study visits, trained study staff conducted interviews of medical history including antiretroviral treatment history, and performed physical examinations (e.g., anthropometry) and phlebotomy.

### Case (incident diabetes mellitus) and control definitions

We defined women as cases with incident, confirmed DM if they met any of the following criteria: a) $\geq$ two fasting blood glucose (FBG) $\geq$126 mg/dL; b) one FBG $\geq$ 126 mg/dL and one random blood glucose (RBG) $\geq$ 200 mg/dL; c) one FBG $\geq$ 126 mg/dL and self-reported DM medications (S1 Table). For each case, the index visit (visit 0) was the visit of DM diagnosis. If participants had two FBG measurements, visit 0 was considered the first date of DM presentation (*i.e.*, first of two DM measurements). All FBG concentrations prior to the index visit were <126 mg/dL. Semiannual visits immediately preceding visit 0 were denoted by the corresponding negative study visit number (*e.g.*, -1 for six months prior, -2 for 12 months prior). We assayed a single stored plasma sample from a study visit between one to two years before the index visit of each case.

We matched every DM case to two controls based on blood glucose, HIV serostatus, use of antiretroviral therapy, race and ethnicity, age ± 15 years, and availability of stored blood sample. To control for metabolic parameters potentially associated with impaired fasting glucose, the first control ("FBG-matched control") was matched on the case's FBG ± 10 mg/dL at the same calendar period visit that their corresponding case had an available stored plasma sample. The second control ("normoglycemic control") had all prior longitudinal glucose values <100 mg/dL and was selected without matching by FBG at the same visit as their corresponding case; this control had a plasma sample available at the same calendar period visit as the case.

### Glucose assays

Fasting blood samples were assayed for glucose concentrations by hexokinase assay (Olympus 5200, 5400 and AU600 automated instruments; Olympus America, Inc., Melville, NY), as previously detailed [22].

## Metabolomic profiling

Plasma samples were collected in sodium citrate (CPT) vacutainers, centrifuged, and stored at -80˚C until thawed for non-targeted metabolomic assays. Plasma samples were randomly sorted by matching strata (DM case, FBG-matched and normoglycemic control) into three sets. Samples in each set were assayed for metabolomic data in a separate run; these three batches are subsequently referred to as WIHS1-3. All sample processing and metabolomic assays were conducted by laboratory technicians blinded to the case or control status of each sample. Initial sample processing to extract metabolites followed the same protocol, which has been previously detailed [23]. Standard operating procedures and quality assurance/quality control of metabolomic assays have also been described [24].

**Liquid chromatography-mass spectrometry.** Plasma samples were assayed in triplicate for metabolomic profiles by time-of-flight liquid chromatography mass spectrometry (LC-MS; Model 6250; Agilent Technologies, Santa Clara, CA) with dual electrospray ionization (ESI) modes [24]. Analytes were separated by C18-based reverse phase column (2 mm x 150 mm Zorbax SB Aq 3.5 um column) in positive and negative ESI modes, which enables greater coverage of features [25]. LC parameters included: autosampler temperature 4˚C, 5 μL injection volume, column temperature 55˚C, and flow rate 0.4 ml/L. The linear gradient was 2–98% of 0.2% (v/v) acetic acid in water (solvent A) to 0.2% (v/v) acetic acid in methanol over 15 min, followed by 2 min hold of solvent B and 5 min post-time. ESI settings included: capillary voltage (Vcap) at 4000 V for positive ion mode and 3500 V for negative ion mode, fragmentor voltage at 135 V, liquid nebulizer at 45 psi, $N_2$ drying gas at 12 L/min and 250˚C. Data were acquired by Agilent MassHunter Qual Workstation Data Acquisition software with the following settings: rate 2.5 spectra/s, centroid mode, and mass scan range 15–2250 [26].

**Metabolomic data extraction and preprocessing.** Each metabolomic feature was defined by a unique mass-to-charge ratio (m/z) and retention time (RT) combination; relative abundance of feature ion intensities were reported as peak areas. An internal reference standard mix included six standard masses ranging from 112.985587 to 1633.949753; this was utilized for mass axis calibration, error assessments and corrections. Major pre-processing steps included: feature detection and extraction; correlation (co-varying ions within each chromatogram); accounting for adducts, isomers, and fragments.

In terms of data-filtering, metabolomic features with ion counts in >80% across participant samples in each data subset (by assay batch [WIHS1-3] and ESI mode [+, -]) were retained for analysis [27]. Missing relative abundance values (e.g. ≤1) were set to the limit of detection (LOD)/2. All feature ion counts were $\log_2$ normalized prior to analysis.

## Statistical and bioinformatic analysis

Analysis was conducted utilizing R (version 4.0.3; R Foundation for Statistical Computing; Vienna, Austria), including MetaboAnalystR [28], and SAS (version 9.4; SAS Institute Inc.; Cary, NC, US). Statistical significance was based on two-sided hypothesis tests, and $\alpha < 0.05$. We initially screened metabolomic features with feature-by-feature unadjusted regressions (Stage 0); since this was a screening criterion, features remained eligible with a $p<0.05$ that was not false discovery rate adjusted. Subsequently, eligible features were evaluated in feature-by-feature adjusted regressions with metabolomic data (Stage 1); false discovery rate (FDR) adjusted q-value <0.05 was considered significant (S2 Fig). We used a complete-case approach for all key variables aside from metabolomic data (S1 Fig).

**Descriptive analysis and visualizations.** Continuous and categorical variables were summarized as medians (interquartile ranges [IQR]) or N's (percentages). Metabolomic features (i.e., $\log_2$ relative abundance) were compared across subgroups by non-parametric test

statistics (e.g. Kruskal-Wallis). $Log_2$-normalized feature relative abundances and clinical indicators were evaluated by Spearman rank-order correlation coefficients. We visually compared differences of $log_2$-normalized feature relative abundances between the three case-control groups via unsupervised dimensionality reduction (principal components analysis [PCA]), supervised discriminant analysis approaches (e.g. partial least squares discriminant analysis [PLS-DA], orthogonal PLS-DA [OPLS-DA]), and hierarchical clustering in heatmaps. Heatmaps were based on calculated Euclidean distances as the similarity index with Ward's linkage as the agglomeration method (clustering based on minimizing sum of squares between any two clusters). We considered permutation test statistics for PLS-DA due to potential overfitting issues.

**Metabolomic feature selection approach.** We utilized a two-stage metabolomic feature selection approach to evaluate the associations between features and case-control status in each data subset (by assay batch [WIHS1-3] and ESI mode [+, -]; (S2 Fig). All conditional logistic models considered a binary categorization of DM cases versus both controls as the primary dependent variable of interest and accounted for matching strata, which reflect individual-matching by blood glucose (FBG-matched, normoglycemic), HIV serostatus, use of antiretroviral therapy, race and ethnicity, age ± 15 years, and availability of stored blood sample. In **Stage 0** screening, unadjusted conditional logistic regressions models assessed the associations between case-control status and $log_2$ feature relative abundance. Metabolomic features differing across groups ($p < 0.05$) were considered eligible for Stage 1 regression models.

In **Stage 1**, multivariable conditional logistic regressions evaluated associations between case-control status and $log_2$ feature relative abundance while accounting for the matching strata and additional covariates. The model equation was:

$$\log\left(p_{DM\ case}/(1 - p_{DM\ case})\right) = \alpha_1 + \alpha_2 z_2 + \cdots + \alpha_S z_S + \beta_0 +$$
$$\beta_1 X_1 \left(\log_2 \text{ feature relative abundance}\right) + \beta_2 X_2 \left(BMI\right) + \beta_3 X_3 (\text{age [years]}), \tag{1}$$

where p = probability of DM case study group, and z = stratum indicator variables (Eq (1)). Metabolomic features were considered associated with the study group (DM cases vs controls) across groups based on $\beta_1$ (FDR-adjusted $q < 0.05$). We only reported Stage 1 results from three lipid classes of interest (fatty acyls, prenol lipids, polyketides), in light of recent lipidomics studies focusing on other lipids classes.

**Feature annotations.** The putative chemical compound identities of metabolomic features were annotated by comparison with lipids curated from METLIN [29]. Annotations were based on monoisotopic accurate mass match (within $\pm\ 10^{-5}$). Selected feature annotations were subsequently manually cross-referenced with Lipid Maps [30] and Human Metabolome Database reference database information [31]. We evaluated feature annotation confidence according to the multi-level system proposed by the Schymanski *et al* [32], which was based on the Metabolomics Standards Initiative (MSI) scoring [33]. Annotations of selected metabolomic features (from adjusted regressions) were considered Levels 2 or 3 [33].

## Ethical conduct of research

The Institutional Review Boards (IRBs) at each WIHS site approved of the study protocol and consent forms (IRB approval numbers: Georgetown University #1993–077, Johns Hopkins University H.34.97.05.19.A2, Montefiore Medical Center #03-07-174, Rush University #13–184, State University of New York Downstate Health Sciences University #266921–64, University of California, San Francisco #21–33925, University of Southern California #

HS-21-00496). All study participants provided written informed consent in English or Spanish prior to voluntary enrollment and data collection.

## Results

One-hundred and fifty women met the inclusion and exclusion criteria and were included in the final analytic dataset. Among these participants, 50 had DM, 50 were FBG-matched controls, and 50 were normoglycemic controls (S1 Fig). Ages ranged from 19 to 62 years at the index study visit; across the three case-control groups, median age ranged from 42 (IQR 36, 48) to 43 (IQR 38, 48; Table 1). In all case-control groups, 80.0% of women had HIV infection (Table 1). Comparing women with HIV infection across the three case-control groups, CD4 cell counts (p = 0.93) and the proportions of women with HIV RNA <400 copies/mL (p = 0.79) were similar (Table 1). Percentages of women on combination antiretroviral therapy

**Table 1. Sociodemographic, clinical, and anthropometric indicators among WIHS participants [a].**

| | DM cases (n = 50) | FBG-matched controls (n = 50) | Normoglycemic controls (n = 50) | p [b] |
|---|---|---|---|---|
| **Sociodemographic** | *Median (IQR) or n (%)* | | | |
| Age (years) | 43.3 (37.5, 47.9) | 42.7 (36.6, 46.4) | 41.8 (35.8, 48.0) | 0.66 [b] |
| Race | | | | |
| White | 12 (24.0) | 12 (24.0) | 12 (24.0) | 1.00 [d] |
| Black | 31 (62.0) | 31 (62.0) | 31 (62.0) | |
| Other | 7 (14.0) | 7 (14.0) | 7 (14.0) | |
| **Clinical** | | | | |
| HIV infection | 40 (80.0) | 40 (80.0) | 40 (80.0) | 1.00 [d] |
| HIV RNA < 400 copies/ml [e] | 18 (45.0) | 16 (40.0) | 15 (37.5) | 0.79 [d] |
| CD4 cell count (cells/mm$^3$) [e] | 476.0 (230.5, 610.0) | 465.5 (238.0, 729.0) | 387.5 (248.5, 646.5) | 0.93 [c] |
| cART [e] | 19 (47.5) | 23 (57.5) | 22 (55.0) | 0.65 [d] |
| Protease inhibitor [e] | 10 (25.0) | 8 (20.0) | 11 (27.5) | 0.72 [d] |
| Stavudine [e] | 8 (20.0) | 8 (20.0) | 7 (17.5) | 0.95 [d] |
| Zidovudine [e] | 12 (30.0) | 13 (32.5) | 11 (27.5) | 0.89 [d] |
| Total # of visits on NRTI [e, f] | 7.5 (1.5, 11.0) | 8.5 (1.0, 11.5) | 6.0 (1.0, 11.0) | 0.96 [c] |
| Family history of DM [g] | 25 (61.0) | 12 (28.6) | 19 (43.2) | 0.01 [d] |
| FBG (mg/dL) | 92.0 (89.0, 104.0) | 93.5 (85.0, 100.0) | 81.0 (76.0, 86.0) | <0.01 [c] |
| HCV infection | 17 (34.0) | 13 (26.0) | 13 (26.0) | 0.60 [d] |
| **Anthropometric** | | | | |
| BMI (kg/m$^2$) [g] | 29.7 (27.6, 36.5) | 28.4 (23.8, 33.5) | 26.0 (22.4, 31.7) | 0.02 [c] |
| Waist circumference (cm) [g] | 97.4 (90.1, 106.5) | 92.4 (82.4, 102.4) | 85.8 (78.7, 98.7) | <0.01 [c] |

[a] At study visit 0 (date of DM diagnosis of cases, and corresponding date of controls in each matching stratum) unless stated otherwise.

[b] Subgroup comparisons based on one-way ANOVA test statistic among continuous variables with normal distribution (Shapiro-Wilk, p>0.05).

[c] Kruskal-Wallis test statistic among non-normally distributed continuous variables (Shapiro-Wilk, p≤0.05).

[d] Likelihood ratio chi-square test statistic among categorical variables.

[e] Only among women with HIV.

[f] Total number of visits from study inception to index visit.

[g] The following covariates were missing among the specified number of participants: Family history of DM (n = 9 cases, n = 8 FBG-matched controls, n = 6 normoglycemic controls), BMI (n = 1 case, n = 2 normoglycemic controls), waist circumference (n = 8 cases, n = 13 FBG-matched controls, n = 7 normoglycemic controls).

Abbreviations: BMI, body mass index; cART, combination antiretroviral therapy; DM, diabetes mellitus; FBG, fasting blood glucose; HIV, human immunodeficiency virus; NRTI, nucleoside reverse transcriptase inhibitor; SD, standard deviation.

(cART), protease inhibitors, stavudine, zidovudine were similar across the three subgroups (all p>0.05; Table 1). Family history of DM was highest among women with DM (61.0%), compared to those in the control subgroups (FBG-matched 28.6%; normoglycemic 43.2%; p = 0.01; Table 1). Median BMI (p = 0.02) and waist circumference (p<0.01) differed across the 3 subgroups (Table 1). Women with DM had the highest median BMI (29.7 kg/m$^2$ [IQR 27.6, 36.5]) and waist circumference (97.4 cm [90.1, 106.5]), compared to the control subgroups (Table 1).

## Comparing relative abundance of metabolomic features by diabetes case and controls status

After data-filtering, 743 metabolomic features remained (S1 and S3 Figs). Stratifying by the six data subsets (based on assay batch [WIHS1-3] and ESI mode [+, -]), the number of remaining metabolomic features ranged between 23 and 273 (S1 and S3 Figs). Considering these metabolomic features in a hierarchical clustering heatmap, the similarity indices (Euclidean distances) appeared distinct across the three case-control groups (WIHS1 participants, positive ESI mode; Fig 1A). Visualizing metabolomic features in each data subset, unsupervised (PCA) and supervised (OPLS-DA) approaches showed similar clustering across the three case-control

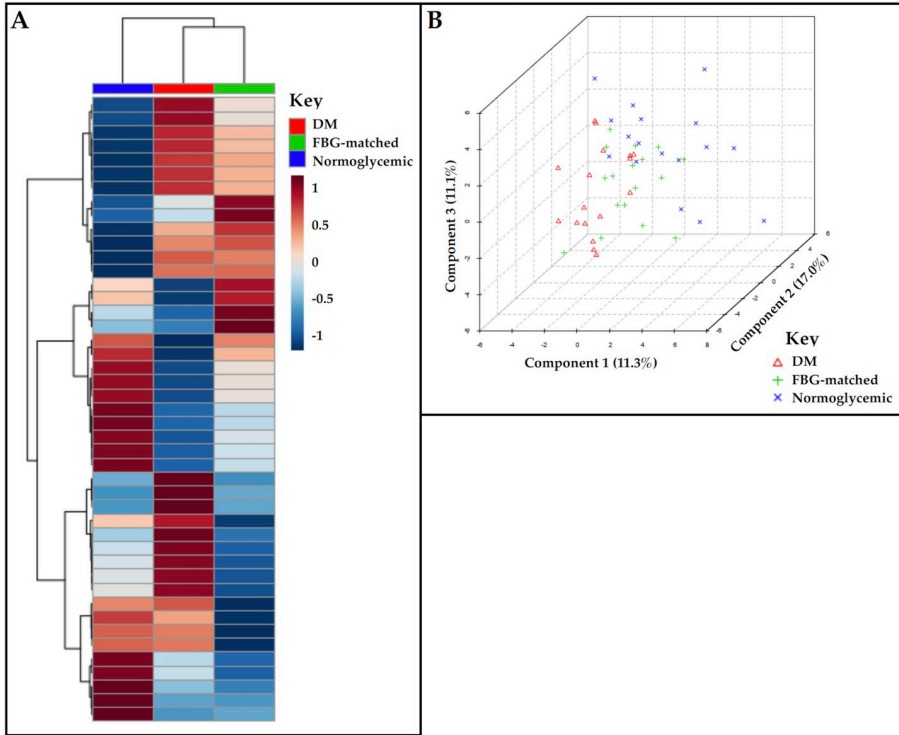

**Fig 1. Comparing metabolomic profiles by DM case and control (FBG-matched, normoglycemic) groups among WIHS 1 participants (n = 51), based on data from C18 (positive ESI). A**: Hierarchical clustering heatmap was based on calculated Euclidean distances as the similarity index with Ward's linkage as the agglomeration method (clustering based on minimizing sum of squares between any two clusters). Log$_2$-normalized relative abundance of metabolomic features are represented in rows; study groups of participants are indicated in columns. DM cases are indicated in red (n = 17), FBG-matched controls in green (n = 17), and normoglycemic controls in blue (n = 17). **B**: Supervised dimensionality reduction was conducted by PLS-DA, in order to visualize clustering across metabolomic features. Study groups are represented as Δ (DM cases), + (FBG-matched controls), and **X** (normoglycemic controls). Abbreviations: DM, diabetes mellitus; ESI, electrospray ionization; FBG, fasting blood glucose; PLS-DA, partial least squares discriminant analysis; WIHS, Women's Interagency HIV Study.

**Table 2. Summary of features differing across DM case and control groups.**

| WIHS discovery, validation sets | | DM case, FBG-matched and normoglycemic controls (# of differing features) | | | | | | Regressions Details | |
| --- | --- | --- | --- | --- | --- | --- | --- | --- | --- |
| | | WIHS1 | | WIHS2 | | WIHS3 | | | |
| **Analytical columns (ESI mode)** | | + | - | + | - | + | - | | |
| N (# of participants) | | 51 | 48 | 48 | 42 | 48 | 51 | - - - | - - - |
| **Feature selection** [c] | | | | | | | | Type | Model equation and details |
| | $N_f$ [b] | 45 | 59 | 273 | 122 | 221 | 23 | | |
| **Stage 0** | p<0.05 | 0 | 3 | 2 | 5 | 13 | 1 | Unadjusted regressions | Conditional logistic regression: log $(p_{\text{DM case}} / (1\text{-}p_{\text{DM case}})) = \alpha_1 + \alpha_2 z_2 + \cdots + \alpha_S z_S + \beta_0 + \beta_1 X_1$ (log$_2$ feature relative abundance), where p = probability of DM case study group, and z = stratum indicator variables |
| **Stage 1** | p<0.05 | 0 | 0 | 2 | 5 | 8 | 1 | Adjusted regressions; among features associated with study group (p<0.05) in unadjusted regressions | Conditional logistic regression:: log $(p_{\text{DM case}} / (1\text{-}p_{\text{DM case}})) = \alpha_1 + \alpha_2 z_2 + \cdots + \alpha_S z_S + \beta_0 + \beta_1 X_1$ (log$_2$ feature relative abundance) + $\beta_2 X_2$ (BMI) + $\beta_3 X_3$ (age [years]), where p = probability of DM case study group, and z = stratum indicator variables |
| | q<0.05 | 0 | 0 | 2 | 5 | 0 | N/A | | |

Abbreviations: BMI, body mass index; DM, diabetes mellitus; ESI, electrospray ionization; FBG, fasting blood glucose; WIHS, Women's Interagency HIV Study.

[a] Values in this table indicate the number of metabolomic features with log$_2$ relative abundance values, which differed by DM case or control (FBG-matched, normoglycemic) group status.

[b] After data filtering, the total number of features considered in each data subset are in S1 Fig. These features were considered via the feature selection approach.

groups (S4 and S5 Figs). Fig 1B shows the first three components from PLS-DA of metabolomic features among WIHS1 participants (positive ESI mode; permutation test statistic p>0.05).

Table 2 summarizes associations between metabolomic features and case-control status (DM cases versus controls), based on unadjusted logistic regressions (**Stage 0**) with conditional likelihood, stratified by data subset. In WIHS1, three metabolomic features (0 in positive ESI mode; 3 in negative ESI mode) were associated with case-control status (all p<0.05). In WIHS2, seven metabolomic features (2 in positive ESI mode; 5 in negative ESI mode) were associated with case-control status (all p<0.05). In WIHS3, 14 metabolomic features (13 in positive ESI mode; 1 in negative ESI mode) were associated with case-control status (all p<0.05).

**Adjusted associations between metabolomic features and diabetogenic subgroups.** In conditional multivariable logistic regressions (**Stage 1**), 7 metabolomic features were respectively associated with case-control status, accounting for matching strata, BMI, and age (all FDR-adjusted q<0.05; Table 2). Per unit increase, two fatty acyls, 6-methyloctan-3one (adjusted odds ratio [aOR] 1.5 [95% CI 1.0, 2.1]; q = 0.04) and sorbic acid (aOR 2.8 [95% CI 1.1, 7.2]; q = 0.04) were associated with elevated odds of diabetes (Table 3). Per unit increases, four polyketides were respectively associated with odds of diabetes, specifically including heteroflavanone C (aOR 0.1 [95% CI <0.1, 0.8]; q = 0.04), rotenonic acid (aOR 0.1 [95% CI <0.1, 0.8]; q = 0.04), louisfieserone A (0.2 [95% CI <0.1, 0.8]; q = 0.04), and (E)-4-nitrostilbene (aOR 1.5 [95% CI 1.0, 2.4]; q = 0.04; Table 3). Podocarpic acid was associated with increased

**Table 3. Associations between selected features and study groups (DM cases versus controls).**

| Lipid category [a] | WIHS data subset [b] | Log$_2$ feature (relative abundance) | | Unadjusted [c] | | | Adjusted [d] | | | | Lipid Maps ID |
|---|---|---|---|---|---|---|---|---|---|---|---|
| | | Variable | Chemical Compound | OR | 95% CI | P [e] | aOR | 95% CI | p [e] | FDR-adjusted q [f] | |
| Fatty acyls | WIHS1 - | | Aminocaproic acid | 4.3 | 1.2, 15.4 | 0.03 | 2.7 | 0.6, 13.0 | 0.20 | 0.20 | LMFA01100035 |
| | WIHS2 - | | 6-Methyloctan-3-one | 1.4 | 1.0, 2.0 | <0.05 | 1.5 | 1.0, 2.1 | 0.04 | 0.04 | LMFA12000129 |
| | | | Sorbic acid | 2.8 | 1.1, 7.1 | 0.03 | 2.8 | 1.1, 7.2 | 0.04 | 0.04 | LMFA01030100 |
| | WIHS3 + | | 3-Oxo-4-methyl-pentanoic acid | 0.6 | 0.4, 0.9 | 0.02 | 0.6 | 0.4, 0.9 | 0.03 | 0.07 | LMFA01020276 |
| | | | 5,11-Dodecadiynoic acid | 0.5 | 0.3, <1.0 | <0.05 | 0.5 | 0.2, <1.0 | 0.04 | 0.07 | LMFA01030464 |
| | | | 10,12-Tetradecadiene-4,6-diynoic acid, (E,E)- | 0.6 | 0.4, 0.9 | 0.02 | 0.5 | 0.3, 0.9 | 0.03 | 0.07 | LMFA01030583 |
| Polyketides | WIHS1 - | | Isosativan | 3 | 1.1, 8.4 | 0.04 | --- [g] | --- [g] | --- g | --- | LMPK12080030 |
| | WIHS2 + | | (E)-4-Nitrostilbene | 2 | 1.1, 3.6 | 0.03 | 1.5 | 1.0, 2.4 | 0.04 | 0.04 | LMPK13090020 |
| | WIHS2 - | | Heteroflavanone C | 0.1 | <0.1, 0.7 | 0.02 | 0.1 | <0.1, 0.8 | 0.03 | 0.04 | LMPK12140478 |
| | | | Rotenonic Acid | 0.1 | <0.1, 0.7 | 0.02 | 0.1 | <0.1, 0.8 | 0.02 | 0.04 | LMPK12060018 |
| | | | Louisfieserone A | 0.2 | <0.1, 0.8 | 0.02 | 0.2 | <0.1, 0.8 | 0.03 | 0.04 | LMPK12140697 |
| Prenol Lipids | WIHS2 + | | Podocarpic acid | 7 | 1.5, 23.7 | 0.01 | 7.1 | 1.5, 33.4 | 0.01 | 0.02 | LMPR0104120002 |
| | WIHS3 + | | Etretinate | 0.2 | 0.1, 0.9 | 0.04 | 0.2 | 0.1, <1.0 | 0.04 | 0.07 | LMPR01090046 |

[a] Lipid categorization per Lipid Maps classification [30]. Features were selected if: 1) associated with case-control status in unadjusted models (p<0.05); and 2) with annotations in lipid classes of interest (fatty acyls, polyketides, prenol lipids).

[b] Data subsets based on metabolomic assay run (WIHS sets 1–3) and ESI mode (+, -).

[c] Unadjusted conditional logistic regression model equation: log (p $_{DM\ case}$ / (1-p $_{DM\ case}$)) = $\alpha_1 + \alpha_2 z_2 + \cdots + \alpha_S z_S + \beta_0 + \beta_1 X_1$ (log$_2$ feature relative abundance), where p = probability of DM case study group, and z = stratum indicator variables.

[d] Adjusted conditional logistic regression model equation: log (p $_{DM\ case}$ / (1-p $_{DM\ case}$)) = $\alpha_1 + \alpha_2 z_2 + \cdots + \alpha_S z_S + \beta_0 + \beta_1 X_1$ (log$_2$ feature relative abundance) + $\beta_2 X_2$ (BMI) + $\beta_3 X_3$ (age [years]), where p = probability of DM case study group, and z = stratum indicator variables.

[e] P value based on Wald chi-square statistic.

[f] Post-hoc FDR adjustment among each data subset (e.g., WIHS1 +) of features evaluated in Stage 1 regressions and with annotations in lipid classes of interest.

[g] Results not reported due to model instability.

Abbreviations: aOR, adjusted odds ratio; DM, diabetes mellitus; ESI, electrospray ionization; OR, odds ratio; WIHS, Women's Interagency HIV Study.

odds of diabetes (aOR 7.1 [95% CI 1.5, 33.4]; q = 0.02; Table 3). Relative abundance of podocarpic acid was compared by case-control status (Fig 2). Data subsets (assay batch [WIHS1-3], ESI mode [+, -]) are specified in Tables 2 and 3.

## Discussion

A total of 743 metabolomic features were observed among participants with DM and their controls matched by blood glucose (FBG-matched, normoglycemic), HIV serostatus, use of antiretroviral therapy, race and ethnicity, age ± 15 years, and availability of stored blood sample. Overall, seven features were significantly associated with odds of DM incidence, accounting for matching strata and after FDR adjustment (all q<0.05). Three flavonoids were associated with lower odds of DM incidence, and sorbic acid was associated with greater odds of DM incidence. Our results indicate the need for confirmation of flavonoids, sorbic acid,

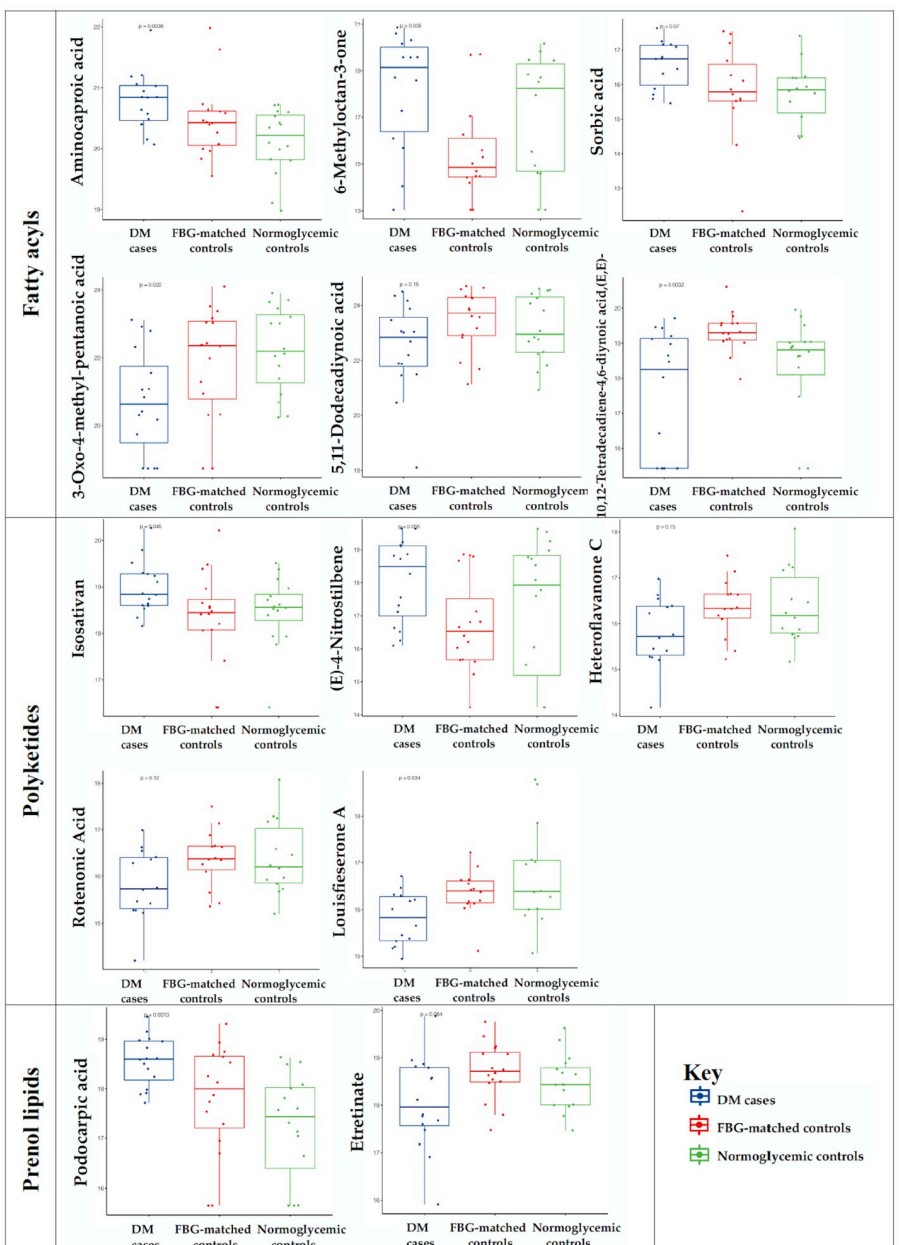

**Fig 2. Boxplots of selected features (relative abundances), stratified by DM case and control groups [a]. [a]** Data subset (e.g. WIHS1 +) specified in Table 3. Abbreviations: DM, diabetes mellitus; FBG, fasting blood glucose.

and their related metabolites via targeted validation with absolute quantitation and mechanistic studies to elucidate their potential respective influences on DM risk.

## Protective effects of flavonoids in diabetes

Phytochemicals synthesized by plants and ubiquitous in the human diet, including many flavonoids [34], are hypothesized to be protective against insulin resistance [35] and DM [36], as well as modulate glucose metabolism [37,38]. Our finding that three flavonoids were associated with lower odds of DM is consistent with the directionality of associations found in

previous studies [36,39], though our exposure assessment was based on circulating metabolites which differs from dietary intake in other studies. In a meta-analysis including 284,806 participants, dietary intake of total flavonoids was associated with lower risk of T2DM [36]. High dietary intake of flavonoids [39] and adherence to plant-based dietary patterns [40] have also been associated with reduced T2DM risk. Prior studies have suggested potential mechanisms to explain this association, including the ability of some individual flavonoids to inhibit oxidative stress [41] and glycogen phosphorylase, which is a primary enzymatic regulator of glucose and glycogen homeostasis [37]. More broadly, polyphenols have been found to affect glucose and insulin metabolism [42], as well as inhibit glycation and advanced glycation end products production [43].

Previous studies have reported mixed associations, including null results, between diabetogenic indicators and dietary supplementation of isoflavones [44,45]. We found that a circulating isoflavan (isosativan) was associated with greater odds of DM, which contrasts with the null or protective associations observed in other observational studies of dietary isoflavonoid intake on DM-related biomarkers [35,45,46]. These inconsistent findings are potentially explained by the unclear mechanisms linking isoflavonoids and DM, which could include mediators and covariates that need to be accounted for (e.g., extensive heterogeneity of DM pathophysiology, observed pleiotropic influences and differing bioavailabilities of isoflavonoids) [34,35,45].

## Elucidating sorbic acid in diabetes

Sorbic acid, or sorbate, is a common synthetic food preservative and metabolite of potassium sorbate, which is a food and pharmaceutical additive [47]. Our finding that sorbate was associated with greater odds of DM is consistent with preliminary evidence of potential explanatory mechanisms [47,48]. Potassium sorbate is completely absorbed after oral ingestion and has cytotoxic and genotoxic influences, which could contribute to elevated risk of a diabetogenic state [47]. Preliminary mechanistic evidence has also shown sorbate to be linked with dysregulated hepatic fatty acid metabolism [48]. Sorbate has also been hypothesized to be an upstream substrate of AGEs [47], which upregulate inflammation and oxidative stress [49] and potentially function as endocrine disrupting chemicals [50]. Future directions of research could examine the: specific metabolic pathways by which sorbic acid and other sorbate additives (*e.g.*, calcium sorbate, potassium sorbate) and other food additives might affect long-term risk of DM incidence, as well as influences of frequency, quantity, timing, and types of sorbates consumed over the human life course on metabolic health.

## Strengths and limitations

A major strength of this study was the nested case-control design within a large ongoing prospective cohort study with standardized protocols [19,20]. Specifically, the study design included the confirmation of each participant with incident DM diagnosis after the measurement of metabolomic features; selection of two individually matched controls based on clinical and sociodemographic criteria; and comparison of stored blood samples collected at the same earlier study visit within each matching stratum. The broad consideration of metabolomic features from non-targeted profiling provided a relatively non-biased perspective. This approach was advantageous given limited prior literature regarding the specific lipid classes of interest in context of DM. Furthermore, the inclusion of only women was a strength in light of sex-based differences in metabolism and DM [51]. Simply controlling for biological sex as a variable in regression models does not preclude residual confounding from other related factors (e.g., sex

hormone differences), since the etiology of many observed sex-linked differences remains incompletely understood [51].

Several limitations should be noted in interpreting results, particularly the modest sample size, inability to determine causal inferences, and single timepoint evaluation of metabolomic data. In the final analysis, we categorized the two control groups into one group, given the sample size per metabolomic assay batch (WIHS1-3). Further validation of metabolites with authentic reference standards and absolute quantification (plasma concentrations) are needed, in order to confirm feature annotations with higher confidence (e.g., Level 1 [32]) and to facilitate comparisons with other populations. We were not able to consider other covariates, such as inflammation, socioeconomic factors, and ART type, and inter-individual variability of gut microbiota [52,53], that potentially influence our associations of interest; future studies should consider these additional covariates. For example, commensal bacteria have been hypothesized to metabolize dietary flavonoids [54] and to be modulated by polyphenols [55] which may subsequently affect metabolic health. Since HIV status was a matching criterion for selecting controls, this study was not designed to evaluate the role of HIV as a comorbidity. However, some flavonoids have antioxidant functions [34] and a recent study demonstrated that two flavonoid glycosides can activate Vδ1+ T cells to suppress HIV-1 [56], emphasizing the need for future studies to consider the associations of individual flavonoids with DM, HIV, and other comorbidities.

## Conclusions

In summary, seven plasma metabolomic features differed among women with DM incidence, compared to their matched controls. Three flavonoids were associated with lower odds of DM incidence. Sorbic acid, a common food preservative, was associated with greater odds of DM. Further studies are needed to validate and delineate the underlying mechanisms of flavonoids and food additives as potential modifiable dietary factors associated with DM, which could improve DM prevention efforts.

## Supporting information

**S1 Fig. Inclusion and exclusion criteria for WIHS study participants, and data filtering of metabolomic features.**
(TIF)

**S2 Fig. Two-stage feature selection approach.**
(TIF)

**S3 Fig. Proportions of feature peak areas observed across participants, stratified by metabolomic assay batch (WIHS1-3) and analytical column (+, - ESI).** In each of the six data subsets, the final analytic subset of participants was considered those individuals in complete matching strata. Features were included below if remaining after data filtering (observed among ≥80% of participant samples).
(TIF)

**S4 Fig. Unsupervised clustering (PCA) of metabolomic features in each data subset (WIHS sets 1–3, positive and negative ESI modes).**
(TIF)

**S5 Fig. Supervised clustering (OPLS-DA) of metabolomic features in each data subset (WIHS sets 1–3, positive and negative ESI modes).**
(TIF)

**S1 Table. Definitions of cases and controls.**
(DOCX)

**S1 File.**
(DOCX)

## Acknowledgments

We thank Kyu Rhee for his assistance with the study design, providing analytical instrumentation, and reviewing the manuscript. The authors gratefully acknowledge the contributions of the study participants and dedication of the staff at the MACS/WIHS Combined Cohort Study sites.

## Author Contributions

**Conceptualization:** Donald R. Hoover, Marshall J. Glesby.

**Data curation:** Elaine A. Yu, José O. Alemán, Donald R. Hoover, Qiuhu Shi, Michael Verano, Kathryn Anastos.

**Formal analysis:** Elaine A. Yu, José O. Alemán, Qiuhu Shi, Michael Verano.

**Funding acquisition:** Kathryn Anastos, Phyllis C. Tien, Mardge H. Cohen, Marshall J. Glesby.

**Investigation:** Michael Verano, Kathryn Anastos, Phyllis C. Tien, Anjali Sharma, Marshall J. Glesby.

**Methodology:** Elaine A. Yu, José O. Alemán, Donald R. Hoover, Qiuhu Shi, Marshall J. Glesby.

**Writing – original draft:** Elaine A. Yu.

**Writing – review & editing:** José O. Alemán, Donald R. Hoover, Qiuhu Shi, Michael Verano, Kathryn Anastos, Phyllis C. Tien, Anjali Sharma, Ani Kardashian, Mardge H. Cohen, Elizabeth T. Golub, Katherine G. Michel, Deborah R. Gustafson, Marshall J. Glesby.

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
