## [Decision Letter · Decision Letter 0]

19 May 2022

PONE-D-21-36113Plasma metabolomic analysis indicates flavonoids and sorbic acid are associated with incident diabetes: A nested case-control study among Women’s Interagency HIV Study participantsPLOS ONE

Dear Dr. Glesby,

Thank you for submitting your manuscript to PLOS ONE. After careful consideration, we feel that it has merit but does not fully meet PLOS ONE’s publication criteria as it currently stands. Therefore, we invite you to submit a revised version of the manuscript that addresses the points raised during the review process.

We look forward to receiving your revised manuscript.

Kind regards,

Anandh Babu Pon Velayutham

Academic Editor

PLOS ONE

Journal Requirements:

This manuscript was also partially supported by NIDDK K08-DK117064 (J.O.A.). 

The WIHS is funded by the National Institute of Allergy and Infectious Diseases (UO1-AI-35004, UO1-AI-31834, UO1-AI-34994, UO1-AI-34989, UO1-AI-34993, and UO1-AI-42590) and by the Eunice Kennedy Shriver National Institute of Child Health and Human Development (UO1-HD-32632). The study is co-funded by the National Cancer Institute, the National Institute on Drug Abuse, and the National Institute on Deafness and Other Communication Disorders. Funding is also provided by the National Center for Research Resources (UCSF-CTSI Grant Number UL1 RR024131). This manuscript was also partially supported by NIDDK K08-DK117064 (J.O.A.). 

Reviewers' comments:

Reviewer's Responses to Questions

**Comments to the Author**

1. Is the manuscript technically sound, and do the data support the conclusions?

Reviewer #1: Partly

Reviewer #2: Yes

2. Has the statistical analysis been performed appropriately and rigorously? 

Reviewer #1: Yes

Reviewer #2: Yes

3. Have the authors made all data underlying the findings in their manuscript fully available?

Reviewer #1: Yes

Reviewer #2: Yes

4. Is the manuscript presented in an intelligible fashion and written in standard English?

Reviewer #1: Yes

Reviewer #2: Yes

5. Review Comments to the Author

Reviewer #1: Abstract

Introduction:

- What is the possible influence of human immunodeficiency virus (HIV) on non-targeted plasma metabolomic profiles of women with incident DM?

- Please, briefly explain why the three lipid classes (fatty acyls, prenol lipids, polyketides) have been chosen.

Methods:

Briefly describe which methods were used to assess metabolomic profile.

Conclusion:

Avoid expressions such as: “further etiological studies are needed”. Focus on the association of flavonoids with lower odds of incident DM and on the association of sorbic acid with greater odds of incident DM.

Main text:

Introduction:

The introduction is still lacking in substance. There have been many more papers about the association between specific food groups and the protective effects against DM incidence.

Methods

Please briefly cite the quality assurance/quality control of metabolomic assays used.

Discussion

- Lines 307-308: The introductory information about flavonoids is not important at this moment (“Flavonoids are phytochemicals synthesized by plants and ubiquitous in the human diet, particularly from many fruits and vegetables”). Focus on the relation between these compounds and DM.

- Lines 356-328: “These inconsistent findings are potentially explained by the unclear mechanisms linking isoflavonoids and DM, which could include mediators and covariates that need to be accounted for”.

Besides, it is important to quote that this is a case-control study, so it is not possible to establish a cause-effect relationship between these factors.

- Line 340: “Future directions of research” … Such as?

- Line 341: “sorbic acid and other food additives”… Such as?

Reviewer #2: This study compared non-targeted plasma metabolomic profiles of women with versus without confirmed incident Type 2 diabetes mellitus (DM). This study focused on examining three lipid classes (fatty acyl, prenol lipids, polyketides). The results showed that two flavanones and one isoflavone, respectively, were significantly associated with a lower incidence of diabetes, and sorbic acid was associated with a higher incidence of diabetes. The paper is well written, and the results are well-presented and interesting.

However, the below points should be considered.

• The studies involving humans should require an Institutional Review Board approval number).

• In addition, informed consent should be obtained from all subjects participating in the study. "

6. PLOS authors have the option to publish the peer review history of their article (what does this mean?). If published, this will include your full peer review and any attached files.

Reviewer #1: No

Reviewer #2: No

---

## [Author Response · Author response to Decision Letter 0]

6 Jun 2022

Academic Editor:

A1. Please ensure that your manuscript meets PLOS ONE's style requirements, including those for file naming. The PLOS ONE style templates can be found at 

Thank you for your comment and feedback.

We have checked the PLOS ONE style templates and finalized our revision materials per the guidelines.

A2. Thank you for stating in your Funding Statement: 

This manuscript was also partially supported by NIDDK K08-DK117064 (J.O.A.). 

We have updated the funding statement with the recommended statement regarding external funding. Please see our response to A3 below.

A3. Thank you for stating the following financial disclosure: 

The WIHS is funded by the National Institute of Allergy and Infectious Diseases (UO1-AI-35004, UO1-AI-31834, UO1-AI-34994, UO1-AI-34989, UO1-AI-34993, and UO1-AI-42590) and by the Eunice Kennedy Shriver National Institute of Child Health and Human Development (UO1-HD-32632). The study is co-funded by the National Cancer Institute, the National Institute on Drug Abuse, and the National Institute on Deafness and Other Communication Disorders. Funding is also provided by the National Center for Research Resources (UCSF-CTSI Grant Number UL1 RR024131). This manuscript was also partially supported by NIDDK K08-DK117064 (J.O.A.). 

We have updated the financial disclosure statement with the suggested sentence.

“The WIHS is funded by the National Institute of Allergy and Infectious Diseases (UO1-AI-35004, UO1-AI-31834, UO1-AI-34994, UO1-AI-34989, UO1-AI-34993, and UO1-AI-42590) and by the Eunice Kennedy Shriver National Institute of Child Health and Human Development (UO1-HD-32632). The study is co-funded by the National Cancer Institute, the National Institute on Drug Abuse, and the National Institute on Deafness and Other Communication Disorders. Funding is also provided by the National Center for Research Resources (UCSF-CTSI Grant Number UL1 RR024131). This manuscript was also partially supported by NIDDK K08-DK117064 (J.O.A.). The funders had no role in study design, data collection and analysis, decision to publish, or preparation of the manuscript. There was no additional external funding received for this study.” 

B. Reviewer 1

B1. Abstract

Introduction:

- What is the possible influence of human immunodeficiency virus (HIV) on non-targeted plasma metabolomic profiles of women with incident DM?

- Please, briefly explain why the three lipid classes (fatty acyls, prenol lipids, polyketides) have been chosen.

Thank you for your constructive comments and suggestions, particularly for the abstract. 

We have revised the abstract, per your comments. Please see our responses below (B2, B3).

We agree that previous evidence has demonstrated links between HIV, antiretroviral therapy, and metabolic abnormalities including diabetes [1-3]. This present study was designed as a nested case-control study with HIV serostatus as a matching criterion, and therefore, we were not able to evaluate the influence of HIV. We have included the need for future studies to evaluate this important research priority in the introduction and discussion (please see text below).

Introduction (Page 4, 26-27): “Diabetes mellitus (DM) is associated with an increasingly heavy burden of disease globally [4, 5], including among people with human immunodeficiency virus (HIV) [6, 7].”

Discussion (Pages 21-22, Lines 372-376): “Since HIV status was a matching criterion for selecting controls, this study was not designed to evaluate the role of HIV as a comorbidity. However, some flavonoids have antioxidant functions [8] and a recent study demonstrated that two flavonoid glycosides can activate Vδ1+ T cells to suppress HIV-1 [54], emphasizing the need for future studies to consider the associations of individual flavonoids with DM, HIV, and other comorbidities.”

Our objective is a separate although complementary question; specifically, we sought to identify low molecular weight chemical compounds associated with the likelihood of diabetes incidence. This was intended to shed light on the key remaining research gaps of: determining specific metabolic pathways that are dysregulated in diabetes pathophysiology; and designing more effective diabetes prevention interventions among people with and without HIV.

We have included the rationale for focusing on three lipid classes in the introduction.

Introduction (Pages 4-5, Lines 47-50): “In a recent study among Women’s Interagency HIV Study (WIHS) participants, cholesteryl esters, diacylglycerols, lysophosphatidylcholines, phosphatidylcholines, and phosphatidylethanolamines were associated with diabetes risk [9].”

Introduction (Page 5, Lines 53-55): “We evaluated lipids and lipid classes that represent potential dietary modifiable risk factors of DM. Specifically, our focus was on three classes of lipids (fatty acyls, prenol lipids, polyketides) [10].”

In order to clearly explain the rationale regarding both of these points, we prefer to discuss these in the introduction and discussion. We did not include these in the abstract as we think it could cause potential confusion if not fully explained.

B2. Methods:

Briefly describe which methods were used to assess metabolomic profile.

We have included further details regarding the metabolomic assays. Our revised text is below.

Abstract, Methods sub-section (Page 3, Lines 11-13): “Stored plasma samples (1-2 years prior to DM diagnosis among cases; at the corresponding timepoint among matched controls) were assayed in triplicate for high-resolution metabolomics. Time-of-flight liquid chromatography mass spectrometry with dual electrospray ionization modes was utilized.”

B3. 

Conclusion:

Avoid expressions such as: “further etiological studies are needed”. Focus on the association of flavonoids with lower odds of incident DM and on the association of sorbic acid with greater odds of incident DM.

We have removed the clause with “further etiological studies are needed” in order to focus the conclusion on key findings from this study (revised text below).

Abstract, Conclusions sub-section (Page 3, Lines 23-24): “Flavonoids were associated with lower odds of incident DM while sorbic acid was associated with greater odds of incident DM.”

B4.

Main text:

Introduction:

The introduction is still lacking in substance. There have been many more papers about the association between specific food groups and the protective effects against DM incidence.

We have updated the introduction, in order to provide more comprehensive context of the previous literature regarding diabetes pathophysiology and the diet. We agree that there is a large literature and therefore cite additional reviews and articles.

Introduction (Page 4, Lines 34-42): “Lifestyle modifications, including healthier dietary patterns with more fruits and vegetables and fewer processed foods, are key prevention recommendations for reducing the risk of T2DM [5]. Despite a large literature regarding specific diets [11] and nutrients [12] in association with diabetes outcomes, findings across some previous studies are inconsistent [13]. It remains a challenge to account for the extensive inter- and intra-individual heterogeneity in consumption patterns, nutritional requirements, dietary responses (e.g., nutrient absorption) [14] as well as the roles of non-nutrients and other dietary components [15]. Evaluation of dietary interventions, particularly long-term adherence, is a major obstacle. Circulating biomarkers of dietary intake could circumvent these issues and potentially serve as improved metrics of specific biologically-active metabolites and earlier predictors of long-term metabolic health [16-18].”

B5. 

Methods

Please briefly cite the quality assurance/quality control of metabolomic assays used.

We have included further details and citations for the quality assurance and control standard operating procedures of the metabolomic assays in this study.

Methods (Page 7, Lines 103-107): “All sample processing and metabolomic assays were conducted by laboratory technicians blinded to the case or control status of each sample. Initial sample processing to extract metabolites followed the same protocol, which has been previously detailed [19]. Standard operating procedures and quality assurance/quality control of metabolomic assays have also been described [20].”

B6.

Discussion

- Lines 307-308: The introductory information about flavonoids is not important at this moment (“Flavonoids are phytochemicals synthesized by plants and ubiquitous in the human diet, particularly from many fruits and vegetables”). Focus on the relation between these compounds and DM.

Thank you for this suggestion. In the sub-section titled “Protective effects of flavonoids in diabetes” (pages 19-20), we have revised this text to focus more on the association between flavonoids and diabetes. Examples of edits include deleting the sentence that Reviewer 1 referenced, as well as the following one, and the first sentence of the subsequent paragraph since these all provided a general introduction to flavonoids and isoflavones.

B7.

- Lines 356-328: “These inconsistent findings are potentially explained by the unclear mechanisms linking isoflavonoids and DM, which could include mediators and covariates that need to be accounted for”.

Besides, it is important to quote that this is a case-control study, so it is not possible to establish a cause-effect relationship between these factors.

Thank you for this comment. We agree that case-control studies do not allow for causal inferences. We included this as a limitation and revised the referenced text for greater clarity (please see below). 

Discussion (Page 21, Lines 362-363): “Several limitations should be noted in interpreting results, particularly the modest sample size, inability to determine causal inferences, and single timepoint evaluation of metabolomic data.”

Discussion (Page 20, Lines 326-332): “We found that a circulating isoflavan (isosativan) was associated with greater odds of DM, which contrasts with the null or protective associations observed in other observational studies of dietary isoflavonoid intake on DM-related biomarkers [21-23]. These inconsistent findings are potentially explained by the need to account for other key mediators and covariates (e.g., extensive heterogeneity of DM pathophysiology, observed pleiotropic influences and differing bioavailabilities of isoflavonoids) [8, 21, 23].”

B8.

- Line 340: “Future directions of research” … Such as?

We have clarified this sentence. Please see the revised text below, which incorporates further details regarding future related research questions.

Discussion (Page 20, Lines 342-346): “Future directions of research could examine the: specific metabolic pathways by which sorbic acid and other sorbate additives (e.g., calcium sorbate, potassium sorbate) and other food additives might affect long-term risk of DM incidence, as well as influences of frequency, quantity, timing, and types of sorbates consumed over the human life course on metabolic health.”

B9.

- Line 341: “sorbic acid and other food additives”… Such as?

Please see our response in B8.

Reviewer 2:

C1. 

This study compared non-targeted plasma metabolomic profiles of women with versus without confirmed incident Type 2 diabetes mellitus (DM). This study focused on examining three lipid classes (fatty acyl, prenol lipids, polyketides). The results showed that two flavanones and one isoflavone, respectively, were significantly associated with a lower incidence of diabetes, and sorbic acid was associated with a higher incidence of diabetes. The paper is well written, and the results are well-presented and interesting.

Thank you very much for your review and suggestions. We have revised the manuscript based on your input.

C2.

However, the below points should be considered.

• The studies involving humans should require an Institutional Review Board approval number).

We have added the Institutional Review Board approval numbers for each of the WIHS clinical sites and the data management center at Johns Hopkins University. Please see the revised text below.

Methods (page 10, lines 188-192): “The Institutional Review Boards (IRBs) at each WIHS site approved of the study protocol and consent forms (IRB approval numbers: Georgetown University #1993-077, Johns Hopkins University H.34.97.05.19.A2, Montefiore Medical Center #03-07-174, Rush University #13-184, State University of New York Downstate Health Sciences University #266921-64, University of California, San Francisco #21-33925, University of Southern California # HS-21-00496). All study participants provided written informed consent in English or Spanish prior to voluntary enrollment and data collection.”

C3.

In addition, informed consent should be obtained from all subjects participating in the study. 

We agree. Please find below the revised text in our response to C2, which states this.

References

1. Koethe JR, Lagathu C, Lake JE, Domingo P, Calmy A, Falutz J, et al. HIV and antiretroviral therapy-related fat alterations. Nature Reviews Disease Primers. 2020;6(1):48. doi: 10.1038/s41572-020-0181-1.

2. Nou E, Lo J, Hadigan C, Grinspoon SK. Pathophysiology and management of cardiovascular disease in patients with HIV. The Lancet Diabetes & Endocrinology. 2016;4(7):598-610. doi: 10.1016/S2213-8587(15)00388-5.

3. Monroe AK, Glesby MJ, Brown TT. Diagnosing and Managing Diabetes in HIV-Infected Patients: Current Concepts. Clinical Infectious Diseases. 2014;60(3):453-62. doi: 10.1093/cid/ciu779.

4. Lin X, Xu Y, Pan X, Xu J, Ding Y, Sun X, et al. Global, regional, and national burden and trend of diabetes in 195 countries and territories: an analysis from 1990 to 2025. Scientific Reports. 2020;10(1):14790. doi: 10.1038/s41598-020-71908-9.

5. World Health Organization. Diabetes. Fact sheet. Geneva: World Health Organization; 2021.

6. American Diabetes Association. 2. Classification and Diagnosis of Diabetes: Standards of Medical Care in Diabetes—2021. Diabetes Care. 2021;44(Supplement 1):S15. doi: 10.2337/dc21-S002.

7. Monroe AK, Glesby MJ, Brown TT. Diagnosing and managing diabetes in HIV-infected patients: current concepts. Clin Infect Dis. 2015;60(3):453-62. Epub 2014/10/15. doi: 10.1093/cid/ciu779. PubMed PMID: 25313249.

8. Manach C, Scalbert A, Morand C, Rémésy C, Jiménez L. Polyphenols: food sources and bioavailability. The American Journal of Clinical Nutrition. 2004;79(5):727-47. doi: 10.1093/ajcn/79.5.727.

9. Zhang E, Chai JC, Deik AA, Hua S, Sharma A, Schneider MF, et al. Plasma Lipidomic Profiles and Risk of Diabetes: 2 Prospective Cohorts of HIV-Infected and HIV-Uninfected Individuals. The Journal of Clinical Endocrinology & Metabolism. 2021;106(4):999-1010. doi: 10.1210/clinem/dgab011.

10. O'Donnell VB, Dennis EA, Wakelam MJO, Subramaniam S. LIPID MAPS: Serving the next generation of lipid researchers with tools, resources, data, and training. Sci Signal. 2019;12(563). Epub 2019/01/10. doi: 10.1126/scisignal.aaw2964. PubMed PMID: 30622195.

11. Sarsangi P, Salehi-Abargouei A, Ebrahimpour-Koujan S, Esmaillzadeh A. Association between Adherence to the Mediterranean Diet and Risk of Type 2 Diabetes: An Updated Systematic Review and Dose-Response Meta-Analysis of Prospective Cohort Studies. Advances in Nutrition. 2022. doi: 10.1093/advances/nmac046.

12. Zheng Y, Li Y, Qi Q, Hruby A, Manson JE, Willett WC, et al. Cumulative consumption of branched-chain amino acids and incidence of type 2 diabetes. Int J Epidemiol. 2016;45(5):1482-92. Epub 2016/07/13. doi: 10.1093/ije/dyw143. PubMed PMID: 27413102.

13. Mustafa ST, Hofer OJ, Harding JE, Wall CR, Crowther CA. Dietary recommendations for women with gestational diabetes mellitus: a systematic review of clinical practice guidelines. Nutrition Reviews. 2021;79(9):988-1021. doi: 10.1093/nutrit/nuab005.

14. Lampe JW, Navarro SL, Hullar MAJ, Shojaie A. Inter-individual differences in response to dietary intervention: integrating omics platforms towards personalised dietary recommendations. Proc Nutr Soc. 2013;72(2):207-18. Epub 2013/02/06. doi: 10.1017/S0029665113000025. PubMed PMID: 23388096.

15. Yates AA, Dwyer JT, Erdman JW, Jr., King JC, Lyle BJ, Schneeman BO, et al. Perspective: Framework for Developing Recommended Intakes of Bioactive Dietary Substances. Advances in Nutrition. 2021;12(4):1087-99. doi: 10.1093/advances/nmab044.

16. Roberts LD, Koulman A, Griffin JL. Towards metabolic biomarkers of insulin resistance and type 2 diabetes: progress from the metabolome. The Lancet Diabetes & Endocrinology. 2014;2(1):65-75. doi: https://doi.org/10.1016/S2213-8587(13)70143-8.

17. Bhupathiraju SN, Hu FB. One (small) step towards precision nutrition by use of metabolomics. The Lancet Diabetes & Endocrinology. 2017;5(3):154-5. doi: 10.1016/S2213-8587(17)30007-4.

18. Rinschen MM, Ivanisevic J, Giera M, Siuzdak G. Identification of bioactive metabolites using activity metabolomics. Nat Rev Mol Cell Biol. 2019;20(6):353-67. doi: 10.1038/s41580-019-0108-4. PubMed PMID: 30814649.

19. Want EJ, O'Maille G, Smith CA, Brandon TR, Uritboonthai W, Qin C, et al. Solvent-dependent metabolite distribution, clustering, and protein extraction for serum profiling with mass spectrometry. Anal Chem. 2006;78(3):743-52. Epub 2006/02/02. doi: 10.1021/ac051312t. PubMed PMID: 16448047.

20. Lakshmanan V, Rhee KY, Wang W, Yu Y, Khafizov K, Fiser A, et al. Metabolomic Analysis of Patient Plasma Yields Evidence of Plant-Like α-Linolenic Acid Metabolism in Plasmodium falciparum. The Journal of Infectious Diseases. 2012;206(2):238-48. doi: 10.1093/infdis/jis339.

21. Cao H, Ou J, Chen L, Zhang Y, Szkudelski T, Delmas D, et al. Dietary polyphenols and type 2 diabetes: Human Study and Clinical Trial. Critical Reviews in Food Science and Nutrition. 2019;59(20):3371-9. doi: 10.1080/10408398.2018.1492900.

22. Rienks J, Barbaresko J, Oluwagbemigun K, Schmid M, Nöthlings U. Polyphenol exposure and risk of type 2 diabetes: dose-response meta-analyses and systematic review of prospective cohort studies. The American Journal of Clinical Nutrition. 2018;108(1):49-61. doi: 10.1093/ajcn/nqy083.

23. Duru KC, Kovaleva EG, Danilova IG, van der Bijl P, Belousova AV. The potential beneficial role of isoflavones in type 2 diabetes mellitus. Nutrition Research. 2018;59:1-15. doi: https://doi.org/10.1016/j.nutres.2018.06.005.

---

## [Decision Letter · Decision Letter 1]

27 Jun 2022

Plasma metabolomic analysis indicates flavonoids and sorbic acid are associated with incident diabetes: A nested case-control study among Women’s Interagency HIV Study participants

PONE-D-21-36113R1

Dear Dr. Glesby,

We’re pleased to inform you that your manuscript has been judged scientifically suitable for publication and will be formally accepted for publication once it meets all outstanding technical requirements.

Kind regards,

Anandh Babu Pon Velayutham

Academic Editor

PLOS ONE

Additional Editor Comments (optional):

Reviewers' comments:

Reviewer's Responses to Questions

**Comments to the Author**

1. If the authors have adequately addressed your comments raised in a previous round of review and you feel that this manuscript is now acceptable for publication, you may indicate that here to bypass the “Comments to the Author” section, enter your conflict of interest statement in the “Confidential to Editor” section, and submit your "Accept" recommendation.

Reviewer #1: All comments have been addressed

Reviewer #2: All comments have been addressed

2. Is the manuscript technically sound, and do the data support the conclusions?

Reviewer #1: Yes

Reviewer #2: Yes

3. Has the statistical analysis been performed appropriately and rigorously? 

Reviewer #1: Yes

Reviewer #2: Yes

4. Have the authors made all data underlying the findings in their manuscript fully available?

Reviewer #1: Yes

Reviewer #2: Yes

5. Is the manuscript presented in an intelligible fashion and written in standard English?

Reviewer #1: Yes

Reviewer #2: Yes

6. Review Comments to the Author

Reviewer #1: (No Response)

Reviewer #2: The authors have adequately addressed the comments I made in the previous round of review, and I feel the manuscript is now ready for publication

7. PLOS authors have the option to publish the peer review history of their article (what does this mean?). If published, this will include your full peer review and any attached files.

Reviewer #1: No

Reviewer #2: No

---

## [Editor Report · Acceptance letter]

30 Jun 2022

PONE-D-21-36113R1 

Plasma metabolomic analysis indicates flavonoids and sorbic acid are associated with incident diabetes: A nested case-control study among Women’s Interagency HIV Study participants 

Dear Dr. Glesby:

I'm pleased to inform you that your manuscript has been deemed suitable for publication in PLOS ONE. Congratulations! Your manuscript is now with our production department. 

Kind regards, 

on behalf of

Dr. Anandh Babu Pon Velayutham 

Academic Editor

PLOS ONE